# Remote Screening for Developmental Language Disorder in Bilingual Children: Preliminary Validation in Spanish–Italian Speaking Preschool Children

Maren Eikerling [1,2], Marco Andreoletti [3], Matteo Secco [4], Bianca Luculli [5], Giulia Cha [5], Sofía Castro [6], Stefania Gazzola [7], Daniela Sarti [7], Franca Garzotto [4], Maria Teresa Guasti [2] and Maria Luisa Lorusso [1,*]

1 Unit of Neuropsychology of Developmental Disorders, Scientific Institute IRCCS E. Medea, 23842 Bosisio Parini, Italy
2 Department of Psychology, Piazza Ateneo Nuovo 1, University Milan-Bicocca, 20126 Milan, Italy
3 Department of Brain and Behavioural Sciences, University of Pavia, Via Agostino Bassi, 21, 27100 Pavia, Italy
4 Department of Electronics, Information and Bioengineering, Politecnico di Milano, Via Ponzio 34/5, 20133 Milan, Italy
5 Department of Speech & Language Therapy "Scientific Institute IRCCS E. Medea", University of Milan, 20122 Milan, Italy
6 Psychology of Language and Bilingualism Laboratory, Institute of Psychology, Jagiellonian University, 30-060 Kraków, Poland
7 Developmental Neurology Unit—Language and Learning Disorders Service, Fondazione IRCCS Istituto Neurologico Carlo Besta, 30060 Milan, Italy
* Correspondence: marialuisa.lorusso@lanostrafamiglia.it

**Abstract:** Due to the difficulties in differentiating bilingual children with Developmental Language Disorder (DLD) from bilingual children with temporary language difficulties that may be caused by heterogeneous language input, language assessments of bilingual children are challenging for clinicians. Research demonstrates that assessments of bilingual children should be in all the languages a bilingual child speaks. This can be arduous for clinicians, but computerised screening approaches provide potential solutions. MuLiMi is a new web-based platform designed to automatise screening procedures for bilingual children at risk of DLD. To validate this procedure and investigate its reliability, 36 Spanish-speaking children, aged 4–6 years old, living in Italy, were tested remotely using the Italian–Spanish MuLiMi DLD screening. Sixteen of the participants were previously diagnosed with DLD. L2 (second or societal language) as well as L1 (first or family language) language abilities in static (nonword repetition, grammaticality judgement, and verb comprehension) as well as dynamic tasks (dynamic novel word learning) were assessed. Speed and accuracy of the children's responses were automatically recorded (except nonword repetition). Significant associations emerged between the results obtained in the screening tasks when comparing them to parental questionnaires and standardised tests. An exploratory analysis of the diagnostic accuracy indicates that the single screening scores as well as the overall total score significantly contribute to DLD (risk) identification.

**Keywords:** bilingualism; Developmental Language Disorder; screening; web-based platform; computerised assessment; remote testing; dynamic novel word learning; nonword repetition

## 1. Introduction

Globally, bilingualism and multiculturalism are considered a widespread and no longer niche phenomenon. Grosjean and Li [1] define bi- and multilingualism as the circumstance in which a person knows or speaks more than one language in everyday life, irrespective of proficiency level in these languages [2]. In the Italian region of Lombardy for example, 11.8% of the migrant population originates from Central and Southern America where Spanish is spoken. Most Spanish-speaking immigrants to Lombardy come from Peru (ca. 4% of the total foreign population in Lombardy, [3]). Globalization as well as migratory

processes have resulted in multilingualism being firmly rooted in everyday life, particularly in educational and SLT contexts [4]. In fact, recent surveys report a high proportion of multilingual children attending SLT services [5]. At the same time, SLTs are not confident in assessing multilingual children or in the service provided to them [6].

Assessing the language abilities of bilingual children continues to be a challenge for clinicians as reported by policy reports [7] and numerous survey studies with Speech and Language Therapists (SLTs), for example [8]. The literature points to the risk of misdiagnoses [9]. Previous research has demonstrated that by using computerised screening solutions, an assessment in both languages spoken is possible, see [10–12]. However, no such instrument is currently available that examines the risk of Developmental Language Disorder (DLD) in Italian-L2 preschool children applying tasks across linguistic domains. We will describe the features and outcomes of the screening platform MuLiMi for the detection of risk of DLD in bilingual preschool children, based on data collected on a group of Spanish–Italian speaking preschool children. The aim of this research is to assess the construct and convergent validity as well as a diagnostic accuracy of screening tasks implemented on the aforementioned newly developed platform.

### 1.1. Background

With a prevalence estimated at around 7% [13,14], DLD is one of the most common developmental disorders. Symptoms of DLD are characterized by difficulties in understanding and/or producing speech or language in communication contexts that occur during childhood [15]. These language problems can refer to all or just some linguistic areas [16]: children with DLD may have problems related to the distinction and production of phonemes (phonetics and phonology) or the processing and realisation of morphemes and syntactic structures (morphology and syntax). Furthermore, the capacity to associate words with meanings as well as to retrieve them can be impaired (semantics and lexicon). Non-verbal communication and the metaphorical use of language can also be problematic in children with DLD (pragmatics). The linguistic deficits may have an impact on the child's general communication abilities and thus affect their social development [17]. Children with DLD are more likely to additionally show deficits in reading and writing acquisition, since poor phonological skills are a risk factor for Developmental Dyslexia [18]. This relates to the potential long-term effects of DLD related to poorer academic performance and lower occupational status [19]. Besides a family history of language impairment (see early evidence for family risk of DLD [20,21]), male gender and a low level of parental education and/or socioeconomic status (SES) are considered risk factors frequently associated with DLD [22–24].

Even though multilingualism does not cause language impairment [25,26] and thus the prevalence for DLD can be expected to be comparable in the mono- and bilingual population [9], associations between language disorders and a migrant background are frequently found [27]. This is likely caused by the heterogeneity of language development in bilingual children, as the two languages do not develop independently from one another. Instead of that, they are interrelated [28,29] and an assessment needs to take such interrelationships into account. Diagnostic procedures that are inappropriate for bilingual children may lead to misdiagnoses [30,31], i.e., over- and underdiagnoses, see [9]. Misdiagnoses carry the risk that insufficient or inappropriate service provision is initiated [32].

Appropriate DLD risk identification of bilingual children—distinguishing between language differences in bilingual acquisitional trajectories vs. clinical conditions—requires assessing the child's language skills in both languages [33,34]. An assessment of both spoken languages has been declared suitable in empirical studies [31,35] and is available for practitioners for certain language combinations. Some assessment tools have been specifically developed for bilingual assessments (e.g., "Prove per la valutazione delle competenze verbali e non verbali in bambini bilingui [Tests for the assessment of verbal and non-verbal skills in bilingual children]" (BaBiL) [36]; "Bilingual English-Spanish Assessment (BESA)" [37]. The LITMUS (Language Impairment Testing in Multilingual Settings)

cross-linguistic lexical tasks (CLTs) [38,39] assesses children's vocabulary skills comparably across languages. In their computerised version (see for example [40] and Zinn, unpublished [41]), CLTs can be automatically administered to bilingual children, which can be beneficial for examiners who typically do not speak all languages spoken by the children under assessment.

When neither knowledge of both languages of the child nor computerised solutions are available, static vocabulary tasks in only one of the languages spoken appear to be insufficient compared to Dynamic Assessment (DA, [42]), which is considered an appropriate and feasible diagnostic tool for DLD in bilingual children [43,44]. DA targets children's language learning potential for the acquisition of new words, applying "test–teach–retest" procedures [45] and systematically providing and recording feedback [46]. In so-called fast mapping tasks for example, an unknown word or nonword (NW, phonotactically legal combination of phonemes without meaning in the language of assessment) is associated with a meaning or object [47], and the child's task is to learn the new association. Such tasks are particularly challenging for children with impaired phonological short term memory and lexical deficits [48]. For bilingual children, some studies point to the particular suitability of fast mapping tasks [49]. Contrastingly, other studies indicate that bilingual TD children also show difficulties in this task [50]. Moreover, consecutive bilingual children seem to perform worse than both monolinguals and simultaneous bilingual children in fast mapping tasks [42]. Kan and Kohnert [51] point to bilingual children's learning potential for such tasks in their L1 (first or family language) being higher than in their L2 (second or societal language).

Besides DA, nonword repetition tasks (NWRTs) involving phonological short-term memory skills have also been shown to be reliable in DLD risk identification in mono- and bilingual children [52] and to be associated with lexical [53,54] and grammatical skills [55]. Children repeat a NW upon auditory presentation and an examiner evaluates the accuracy of the child's repetition. Repetition of the meaningless NWs is less influenced by language exposure patterns than word or sentence repetition and is thus appropriate for application with bilingual children [56]. Still, the degree of language specificity of the NWs should be taken into consideration [12].

In addition to that, language-specific clinical markers and grammatical features, in particular, can be used to differentiate between the linguistic profiles of monolingual children with DLD and those of multilingual TD children [7]. In Italian, the omission of articles and clitic pronouns as well as the replacement of the third person plural form of the verb (e.g., cantano, 'they sing') with the third person singular form (e.g., canta, 'she/he sings') are considered clinical markers of DLD [57–60] in both monolingual and bilingual populations [61,62]. In Spanish, errors in subject–verb agreement concerning first person forms replaced with third person forms are considered a clinical marker [63]. Furthermore, Grinstead and colleagues [64] found the use of infinitive verbs instead of inflected verbs to be an indicator of DLD.

In the last decade, web-based language testing has been more extensively used with second language learners due to increased reliability and flexibility [65]. However, there have been limitations to the use of web-based language testing, primarily being the constraints in the digital literacy of the target population [66]. Even though to a lesser extent, in clinical contexts, the application of computerised language assessment is also increasing, see for example [67]. There are many new developments in web-based language testing assessments such as the previously mentioned BaBiL [36] and German "Evozierte Diagnostik grammatischer Fähigkeiten für mehrsprachige Kinder [Evoked diagnostics of grammatical skills for multilingual children]" (ESGRAF-MK, [68]). These web-based language assessments allow for the automatic administration of tasks in the child's societal language (Italian in the case of the BaBiL [36] and German in the case of the ESGRAF-MK [68]) and a selection of first/family languages. Upon the computer-assisted administration, examiners evaluate the child's responses in their first/family language following indications on linguistic structures and short descriptions of the target answers in the manual. In addition to

automatic administration, other solutions provide automatic evaluation, either of dynamic tasks, see [69,70], or of static tasks in both languages spoken by a bilingual child (see [40] and Zinn, unpublished [41]).

Currently, the demands on static tasks related to (a) different language combinations (for example, resulting from current migration flows), and (b) increasing scientific knowledge regarding the suitability of certain task contents and types, are steadily evolving. On this ground, the modifiable screening platform MuLiMi was created within the framework of the multi-centred, interdisciplinary, EU-funded research project MultiMind—The Multilingual Mind. It allows for the construction and automatic administration of reading [11] and language screening tasks [10], as well as for the evaluation of children's responses in various languages.

*1.2. Rationale and Research Goals*

The literature review shows that the identification of DLD in multilingual children is challenging but also possible from a (cross-)linguistic perspective. This implies that not only are computerised language assessments available in various languages, but they are also language-specific, i.e., they take into consideration specific and disparate markers of DLD for each of the languages assessed. Furthermore, research has shown that computer-based methods are efficient in assessing the same, even remote-based, methods. No previous research studies known to the researchers have yet applied these resources to identify DLD among Spanish-speaking preschool children in Italy. The present screening tasks were developed to automatise the bilingual screening of preschool children from migrant backgrounds living in Italy.

This research aimed to preliminarily validate the remote screening procedure and tests its applicability for the detection of DLD among bilingual preschool children from migrant backgrounds. As mentioned above, Spanish is among the most common languages of the population with a migration background in the Lombardy region. More specifically, the data obtained in the sample of Spanish–Italian speaking children will be analysed in terms of discriminant and convergent validity as well as diagnostic accuracy:

1.  Construct and convergent validity will be assessed through the correlations between standardised and screening tests and questionnaires assessing the same or similar abilities.
2.  The diagnostic accuracy of the experimental battery will be exploratorily investigated through the analysis of the different receiver operating characteristic (ROC) curves. With ROC curves, we will distinguish the clinical condition vs. typical language acquisition, allowing for the definition of ideal cut-off points for the single tasks. To this end, the Youden index (J) will be considered. The parameters describing the ROC curves of the total screening battery as well as those of the single tasks will be taken as indexes of discriminant validity.

## 2. Materials and Methods

*2.1. Participants*

Thirty-six early-sequential or simultaneous bilingual Spanish–Italian-speaking children aged between 49 and 76 months (mean age in months: $M = 64.50$, $SD = 7.87$), $n = 16$ female (44.4%) and $n = 20$ male (55.6%) participated in this study. Sixteen of these children had already been diagnosed with DLD by a specialized professional and received SLT treatment. Nine children neither held a DLD diagnosis nor did they score below the cut-off in the standardised tests and were thus considered typically developing (TD). Another 11 children had not been diagnosed with DLD, but scored below the cut-off according to the standardised test manuals, i.e., two standard deviations (SD) below the mean on at least one test, and were thus considered at risk of DLD. Accordingly, two variables were created to describe the child's status: (1) a three-level variable referred to as "risk level" (DLD, at-risk, and TD), (2) a dichotomous variable indicating the presence vs. absence of either a DLD diagnosis or DLD risk. Children with known neurological or intellectual

disorders were excluded. All children lived in Italy, attended kindergarten, and had been continuously exposed to the Italian language for at least two years. At least one of the child's parents or caregivers was a native speaker of Spanish. The recruitment of children took place in private and public kindergartens as well as private and public clinical centres providing SLT services in the Lombardy region, Italy. According to a questionnaire filled in by the children's caregivers (QUIR-DC, see Section 2.4), all children were exposed to both languages on a daily basis, except for three children who, in addition to Spanish and Italian, were also exposed to English ($n$ = 2) and Somali ($n$ = 1). The children in the TD group, on average, were more exposed to Spanish and used it more frequently than the other two groups. More precisely, 75% of TD children, 50% of children from the at-risk group, and 21% of the children with DLD received input regularly in Spanish from at least one of their parents. In total, 75% of TD children, 30% of the at-risk group, and 31% of the children with DLD used Spanish frequently.

Demographic data for the individual subgroups were (a) DLD: mean age in months $M$ = 66.63, $n$ = 10 female, and $n$ = 6 male; (b) at-risk: mean age in months $M$ = 60.64, $n$ = 5 female, and $n$ = 6 male; (c) TD: mean age in months $M$ = 65.44, $n$ = 1 female, and $n$ = 8 male. Chi$^2$ analysis revealed that the distribution of gender was different across the groups, with a higher presence of females in the DLD group versus the other groups (chi$^2$ = 6.167, $p$ = 0.046).

As to SES-related variables, the educational level of the mothers was (a) DLD: 4 middle school, 10 high school, and 2 master's degree; (b) at-risk: 4 middle school, 4 high school, and 2 master's degree; (c) TD: 3 middle school, 1 high school, and 4 master's degree. The educational level of the fathers was (a) DLD: 12 middle school, 3 high school, 1 master's degree; (b) at-risk: 6 middle school, 2 high school, 1 master's degree; (c) TD: 2 middle school, 2 high school, 4 master's degree. In neither case did the chi$^2$ analysis show significantly different distributions across the groups.

### 2.2. MuLiMi Platform and Tasks

The Spanish–Italian screening targets the assessment of language skills in the family and societal language, thus consisting of similar task types and language-specific clinical markers where possible in both Spanish and Italian across linguistic areas. All screening tasks were implemented on and administered through the MuLiMi platform, a web platform developed by the Scientific Institute "E. Medea" with the support of Politecnico di Milano to allow researchers to independently customise the tasks (see [11]). This means that an administrator can upload audios, pictures, and videos to the platform, modify them according to linguistic or cultural characteristics of the target group, and merge these contents according to the target structure. As described by Eikerling and colleagues [11], examiners can administer the screening remotely through link share, enabling (upon consent) a screen share of the examinee interface with the examiner's device. For all but the nonword repetition task (see Section 2.2.1) and the dynamic novel word learning naming phase (see Section 2.2.4), the examinee's performance on the tasks (speed and accuracy) is scored automatically. It can be viewed and exported by the examiner in the MuLiMi interface.

A description of each of the tasks implemented on the platform is reported below.

### 2.2.1. Nonword Repetition Task

For information on the methodology applied for the construction of the nonword repetition task (NWRT, i.e., the inclusion or exclusion of language-specific phoneme inventories, and the recording by native speakers with native-like vs. neutral intonation), a selection of NWs (two-step rating L1 of alikeness and pronounceability) as well as the automatic administration (computer-assisted presentation of pre-recorded nonwords and visual feedback), and manual evaluation of the child's repetition performance (correct vs. incorrect repetition of a nonword), see [12]. For the selected set of nonwords, inter-rater-reliability was $\alpha \geq 0.602$ and internal consistency was $\alpha$ = 0.747. The list of NWs used in the Spanish–Italian screening consists of a total of 10 NWs, precisely, $n$ = 4 Italian language-specific

nonwords (LS IT, e.g., [blan'djeza]), *n* = 2 non language-specific NWs spoken by a native Italian speaker (NLS, e.g., [fulsαmit]), and *n* = 4 Spanish language-specific NWs (LS SP, e.g., [ajukom'jon]), see Table A1. Syllable length varies from two to four syllables and is fairly distributed among the four categories. In addition to the procedure analogous to Eikerling and colleagues [12], an instruction video by a Spanish native speaker was presented. All *n* = 10 LS IT, NLS, and LS SP NWs were automatically presented in a random order using the MuLiMi screening platform and repeated by the child one after another. Repetition performance (correct 1 point vs. incorrect 0 points) was manually scored. In order to reflect the children's specific language acquisition conditions acquiring both Italian and Spanish, the scores used were based on the mean score of the Spanish- and an Italian-speaking rater.

### 2.2.2. Cross-Linguistic Lexical Tasks

The verb comprehension subtests in Italian and Spanish from the cross-linguistic lexical tasks (CLTs, [38,39])—provided by the authors for research purposes—were implemented on the MuLiMi screening platform for automatic administration and scoring (accuracy) of the children's performance. The children's task was to select the picture out of four that corresponds to the auditorily presented verb. The Italian audio files for instructions and item presentation were the same ones that were used in the CLT app (Zinn, unpublished, see [41]). For Spanish instead, the instruction and items were recorded by a native speaker of Spanish as indicated in the Spanish version of the CLTs (Cantù Sanchez et al., unpublished). Each of the subtests in both languages contains 32 items (64 in total). The accuracy of the responses was automatically measured and stored.

### 2.2.3. Who Says It Right?

In the "Who Says It Right?" (WSIR) paradigm, the pre-recorded question for grammaticality judgment "Who says it right?" is first presented auditorily in the target language. Then, one grammatically correct and one grammatically incorrect sentence (see types of manipulation in Table A2) are presented auditorily in a random order while two different figures (GIF-files) that seem to be saying each one of the sentences are presented on the screen. In addition, a coloured line drawing depicting the scene described in the sentence is displayed (see Figure 1).

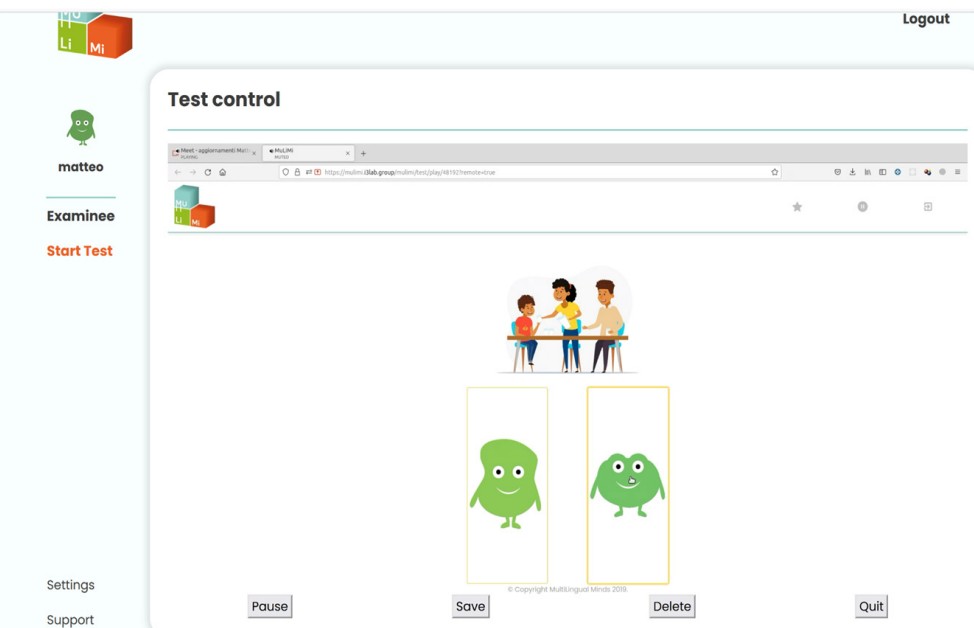

**Figure 1.** Examiner interface during remote administration of the WSIR subtest.

The child is asked to indicate which of the two sentences presented is correct by selecting the figure that appeared to be pronouncing it. Two types of WSIR tasks are used in both languages. One of the tasks compares correct and incorrect sentences manipulated for subject–verb agreement (SVA) and the other targets sentences manipulated for the presence/absence of (incorrect) infinitive forms of the verbs (finiteness, FIN) (see examples in Table A2). For either of the language versions of this task, an instruction video with a native speaker explaining the task is presented prior to task completion. While each language version of the WSIR SVA task consists of 2 training and a total of 16 screening items, each language version of the WSIR FIN task consists of 2 training and a total of 8 screening items. Accuracy of the given responses are automatically measured and stored. Due to time constraints, the WSIR FIN task was not administered to all children.

### 2.2.4. Dynamic Novel Word Learning

In the dynamic novel word learning (DNWL) task, three auditorily presented LS NWs (for procedure of construction and selection, see [12]; for the Italian version [galpo], [dɔmjo], and [feːljo]; for the Spanish task [mokal], [flado], and [leɲon]) were associated with one character displayed on the screen that did not resemble existing creatures or objects (see Figure 2). It was carefully avoided to use words that could easily be associated with any distinctive feature of the characters. This task involves four phases: presentation, consolidation, test, and naming phase. Task instructions and stimuli were recorded by native speakers.

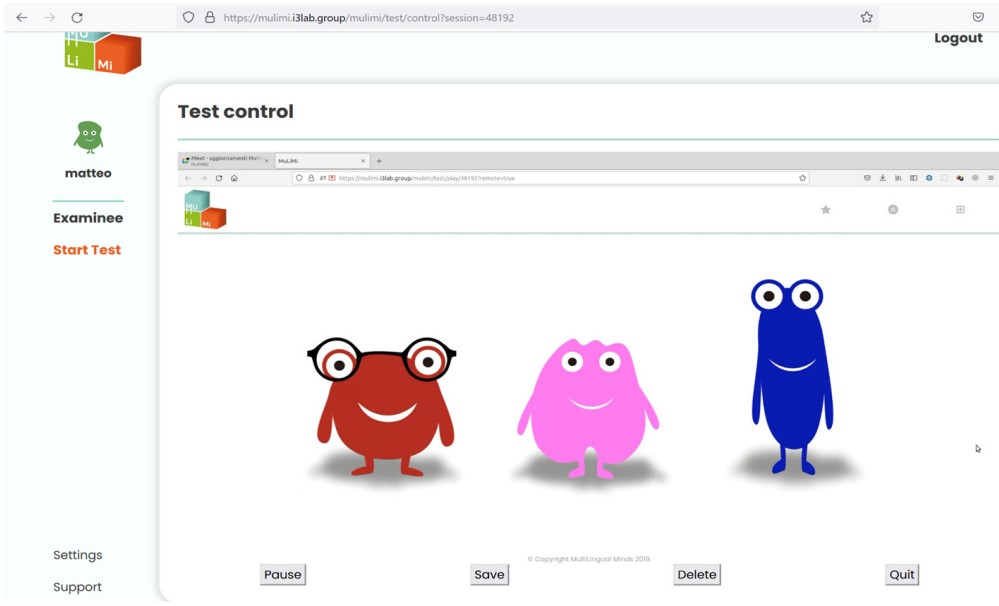

**Figure 2.** Examiner interface during remote administration of the DNWL consolidation phase.

In the **presentation phase**, the characters are displayed one by one on the screen. Upon the auditory presentation of the corresponding "name", the child is asked (by a pre-recorded audio by native speakers of the language of assessment) to click on the character that was just introduced to encourage their active engagement with this task. Corrective feedback is provided upon an incorrect selection, but responses are neither collected nor analysed.

In the **consolidation phase**, all characters were displayed on the screen and children were asked to select each character one by one when it was named. When selecting the target at first attempt, corroborative, pre-recorded auditory feedback is provided and the next name is presented. Having selected the wrong character instead, upon auditory corrective feedback, the child is asked the same question again until the correct figure has been identified. The system uses two variables to manage this phase: $c$ and $c_{err}$, tracking,

respectively, the next character to be presented and the number of errors in the current execution of the phase. After all the characters have been presented, if $c_{err}$ is greater than zero, the phase is repeated (see Figure 3). The frequency of correct character identification along with the total number of attempts is automatically recorded, and the ratio is derived and recorded.

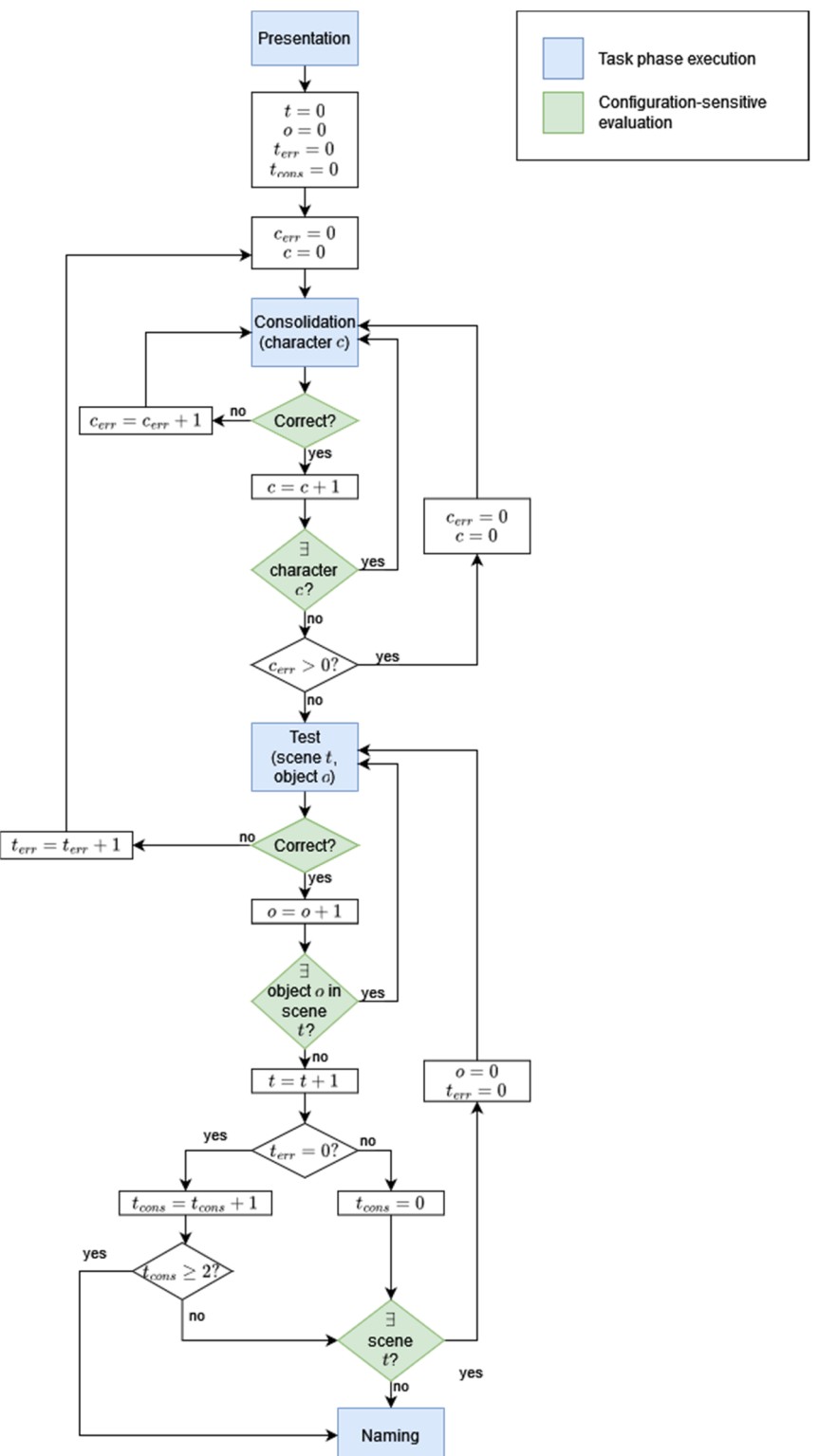

**Figure 3.** Flow diagram of the DNWL task.

In the **test phase**, the NWs associated with characters are embedded within semantic contexts, visually supported by coloured line drawings depicting the scene. In these scenes, each character requests an object (high-frequent nouns, selection based on CDI database) [71]. A pre-recorded audio is played introducing the scene (e.g., "Now they go out."), the condition of a certain character is described (e.g., "Felio notices the rain."), and a question is posed that requires the association between the information in the scene and the character's name (e.g., "Whom do you give the umbrella to?"). Upon the child selecting the corresponding character, a subtle audio signal indicates that the answer has been recorded, irrespective of the accuracy of the child's response. This subtest consists of a minimum of two and maximum of three scenes, depending on the child's performance. The state of this phase is represented by four variables: $t$ and $o$ represent the current scene and the current object within the scene, $t_{err}$ counts the number of incorrect answers given within the current scene, and $t_{cons}$ counts the number of consecutive scenes completed with no errors before the current one. If in the first or second scene the participant answers incorrectly to one or more of the questions proposed, the consolidation phase is repeated. The test phase resumes from the mistaken object. If the examinee provides exclusively correct answers in the first two scenes (having $t_{cons}$ reaching a value of 2), the phase ends skipping the third scene (see Figure 3). Again, the frequency of correct character identification and the amount of attempts are automatically recorded. In each scene, all three characters are addressed once in a random order, so this phase consists of a maximum $n = 9$ items.

Finally, in the **naming phase,** the context of the characters is introduced. The child is asked to call the characters one by one. If the child does not answer after 5 s or declares not to know the answer, he/she is cued with the initial sound of the character's name. Children's answers are evaluated manually by the examiner (see Table A3).

### 2.3. Standardised Tests

To validate the screening tasks, subtests from common standardised language tests for Italian-speaking preschool children were used to assess performance in the participants' societal language. The scores were then used to create risk scores in single linguistic domains.

Children were administered four subtests of the "Batteria per la Valutazione del Linguaggio in Bambini dai 4 ai 12 anni" (BVL, [72]): nonword repetition to assess phonological skills, sentence repetition, grammaticality judgement, and sentence completion subtests to assess morphosyntactic skills. To further assess the children's lexical skills in their societal language Italian, the vocabulary comprehension subtests of the "Test Fono-Lessicale: Valutazione delle abilità lessicali in età prescolare" (TFL, [73]) were used. As recommended in the context of language assessment for multilingual children, only the results of the Italian screening task on verb comprehension were compared to the Italian standardised TFL [73]. Nonverbal abilities were tested using the "CPM-Coloured Progressive Matrices" [74], under the supervision of a trained psychologist, if the scores of standardised nonverbal intelligence tests were not retrievable from clinical records or were more than one year old. All the results of the standardised tests were evaluated according to published manuals and norms. Raw scores were converted into percentages to facilitate comparison with the results of the experimental tests.

### 2.4. Questionnaires [75]

Kindergarten teachers or SLTs judged the abilities of each child in the domains of phonology, lexicon, morphosyntax, and pragmatics on a four-level scale (no difficulties, mild/moderate/severe difficulties for receptive and productive skills, where applicable). From these single scores, the sum was calculated and referred to as "observed difficulties".

Caregivers chose between the online and pen-and-paper version of the QUIR-DC scale [75]. The questionnaire consists of $n = 111$ questions with multiple answers (and a few open answers) and it is automatically scored. The caregivers are requested to provide anamnestic, social, and family information that has been linked to the risk of

language disorders, to describe the child's cognitive and linguistic past and present behaviours that can be considered as potential markers of the presence of cognitive, linguistic, or communication disorders. The questionnaire is still in the process of being validated and no validity or reliability scores are available at present. Its application with monolingual children has proved useful for the distinction of typically developing, language-impaired, and autistic children [75,76]. The caregivers' answers to the QUIR-DC questionnaire are subsumed into the global score (GS) expressing the level of development. Negative scores contribute to the risk score (RS) expressing the risk of a developmental delay or disorder. Caregivers additionally filled in a short section on the language background of the family (e.g., language(s) spoken by the caregivers, languages spoken by the child, time of residence in Italy, and notion of differences in the languages spoken by the child).

### 2.5. Procedure

Recruitment and data collection took place between September 2020 and November 2021 with breaks due to COVID-19-related restrictions in accessing schools and clinical centres, where SLT diagnoses and interventions take place. All children were tested individually in the SLT clinic or kindergarten where they were recruited, in the presence of their SLT, kindergarten teacher, and/or a student researcher in a quiet room. In the first of two testing sessions, the trained student researchers or an SLT (native Italian speakers) administered the standardised tests which lasted around 45 to 60 min. In the second session, they took on the role of a "supporter": the examiner was connected with the supporter and the child participant via a conference call from remote and administered screening tasks through link share mechanisms provided by the MuLiMi screening platform, see [11]. For the simulation of a realistic testing scenario, the child was connected to the examiner using the laptops or desktop PCs available to the student researcher or in the clinic/kindergarten where they were tested. Due to technological constraints and comparability of screen size for the screening and standardised test visualisation, participation in the study from tablets or smartphones was not possible. The supporter assisted in establishing the connection to the screening's examiner and recorded the child's responses to the NWRT and naming phase in the DNWL on a portable recording device. When the device available did not allow for reacting to the stimuli presented via a touchscreen and the child was not familiar with using the mouse, he/she was instructed to indicate the button to click with the index finger and the supporter clicked on the button correspondingly. The heterogeneity in responding to the stimuli was considered acceptable, since not response time but accuracy is considered in this step of the screening validation. This session lasted between 45 and 60 min. A break of minimum 60 min separated the first and the second testing session. Whenever necessary, the child took a short break.

### 2.6. Data Analysis

Considering that both a diagnosis based on the application of monolingual tests and the absence of a diagnosis in a population with a migration history cannot be considered reliable, a simple distinction between children with and without a DLD diagnosis could not provide a solid and sufficient benchmark for validation. Therefore, the results obtained in the standardised language tests were used to identify children at risk of DLD (below the cut-off in at least one of the standardised tests) and children without DLD (standardised test scores above the cut-off and no DLD diagnosis, see Section 2.1, Participants section). Moreover, to provide an accurate validation of each of the battery subtests (assessing lexical, phonological, or morphosyntactic abilities), a further, more specific criterion for classifying children at risk was applied for the various linguistic areas assessed individually. The participants were divided into two groups (risk/no risk) based on the specific clinical cut-offs described in the respective standardised tests' manual. Thus, the at-risk group consisted of those children without a formal diagnosis of DLD, but who showed at least one score significantly below the norm on standardised

tests in Italian, according to the manual of the respective test. The information on whether the child had scored below cut-off in the TFL comprehension subtest determined the presence or absence for the lexical risk (children with lexical risk scored $\leq$ 5th percentile: $n$ = 5), the BVL scores in the subtests on sentence completion and repetition determined the morphosyntactic risk (children with a morphosyntactic risk score $\geq$ 2 SD: $n$ = 14), and the phonological risk was based on the BVL nonword repetition task (children with a morphosyntactic risk score $\geq$ 2 SD: $n$ = 19).

With the aim of assessing the overall capacity of the screening in identifying the risk of DLD, three different scores were created: (1) the Italian overall score (ITscore) which consists of the weighted average of the percentages of correct answers in the Italian tasks in the different linguistic areas (lexical, phonological, and morphosyntactic); (2) the Spanish overall score (SPscore) which consists of the balanced average of the percentages of correct answers in the Spanish tasks in the different linguistic areas; and (3) the bilingual overall score (BILscore), i.e., weighted average of the percentages of correct answers in the whole linguistic areas in the two different languages. More precisely, since in both languages there are two tasks assessing morphological skills (WSIR agreement and finiteness), the weighted average of the percentages of correct answers in both morphology tasks was calculated before the creation of the overall scores (i.e., each of the two tasks was assigned a weight of 0.5), so that each language area contributes equally to the final score.

## 3. Results

None of the screening task results were significantly associated with the children's test scores in the nonverbal intelligence tests ($p_s$ > 0.05); therefore, this variable was not taken into account as a control variable in the analyses.

### 3.1. Overall Screening Score

The diagnostic accuracy of the three different overall scores (Italian overall score, Spanish overall score, and bilingual overall score) was exploratorily assessed with respect to (a) the presence of a DLD diagnosis vs. the absence of pathological scores in any of the Italian standardised tests and (b) the presence of a DLD diagnosis and/or pathological scores in any of the Italian standardised tests vs. the absence of any pathological scores (thus, merging the DLD group and the risk group).

Unfortunately, due to technical or personal motivation or collaboration issues, some children had sporadic missing data on one or more of the tests. Therefore, the overall score could not be computed for all the children, and the number of data available for the Italian, for the Spanish, and for the bilingual overall scores are not identical.

Diagnostic accuracy was analysed with respect to the **presence of a DLD diagnosis** as opposed to the absence of pathological scores. For the Italian overall score, data of a total of $n$ = 16 children were analysed, including 8 with and 8 without DLD, obtaining an AUC value of 0.938, $p$ < 0.001. A cut-off point of 60.42% correct responses would allow for a sensitivity of 0.750 and a specificity of 1.000 ($J$ = 0.750, see Figure 4a). For the Spanish overall score, data of a total of $n$ = 16 children were analysed, including 9 with and 7 without DLD, obtaining an AUC value of 0.873, $p$ < 0.001. A cut-off point of 44.79% correct responses would allow for a sensitivity of 0.667 and a specificity of 1.000 ($J$ = 0.667, see Figure 4b). For the bilingual overall score, data of a total of $n$ = 13 children were analysed, including 7 with and 6 without DLD, obtaining an AUC value of 0.905, $p$ < 0.001. A cut-off point of 57.4% correct responses would allow for a sensitivity of 0.714 and a specificity of 1.000 ($J$ = 0.714, see Figure 4c).

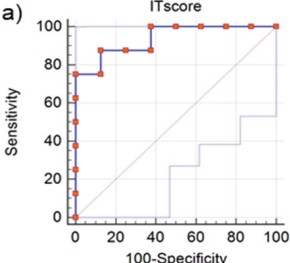 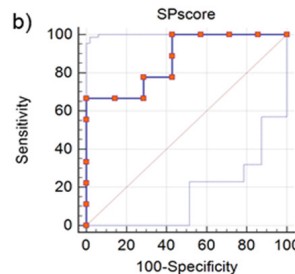 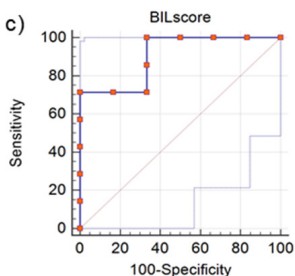

**Figure 4.** (**a**) ROC curve for the Italian overall score (distinction between DLD diagnosis, *n* = 8, and absence of pathological scores, *n* = 8), AUC = 0.938; (**b**) ROC curve for the Spanish overall score (distinction between DLD diagnosis, *n* = 9, and absence of pathological scores, *n* = 7), AUC = 0.873; (**c**) ROC curve for the bilingual overall score (distinction between DLD diagnosis, *n* = 7, and absence of pathological scores, *n* = 6), AUC = 0.905.

Diagnostic accuracy was further analysed with respect to the **presence of a DLD diagnosis and/or pathological scores on Italian standardised tests** (DLD and risk groups) as opposed to the absence of such conditions (TD). For the Italian overall score, data of a total of *n* = 24 children were analysed, including 8 TD and 16 with DLD diagnosis and/or pathological scores, producing an AUC value of 0.969, *p* < 0.001. A cut-off point of 63.19% correct responses would allow for a sensitivity of 0.875 and a specificity of 1.000 (*J* = 0.875, see Figure 5a). For the Spanish overall score, data of a total of *n* = 23 children were analysed, including 7 TD and 16 with DLD or pathological scores in standardised tests, obtaining an AUC value of 0.857, *p* < 0.001. A cut-off point of 64.58% correct responses would allow for a sensitivity of 1.000 and a specificity of 0.571 (*J* = 0.571, see Figure 5b). For the bilingual overall score, data of a total of *n* = 19 children were analysed, including 6 TD and 13 with DLD diagnosis and/or pathological score, producing an AUC value of 0.910, *p* < 0.001). A cut-off point of 57.40% correct responses would allow for a sensitivity of 0.714 and a specificity of 1.000 (*J* = 0.714, see Figure 5c).

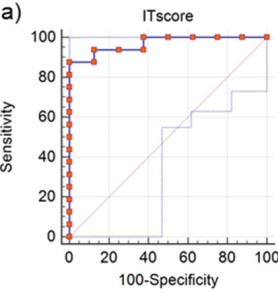 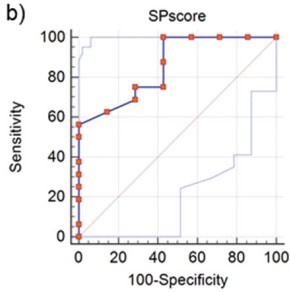 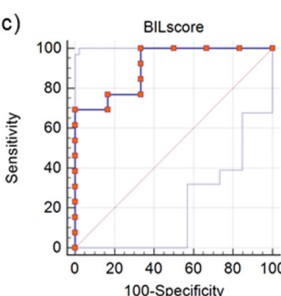

**Figure 5.** (**a**) ROC curve for the Italian overall score (distinction between presence (*n* = 16) vs. absence of pathological scores or DLD diagnosis, *n* = 8), AUC = 0.969; (**b**) ROC curve for the Spanish overall score for the presence of DLD and DLD risk (distinction between presence (*n* = 16) vs. absence of pathological scores or DLD diagnosis, *n* = 7), AUC = 0.857; (**c**) ROC curve for the bilingual overall score (distinction between presence (*n* = 13) vs. absence of pathological scores or DLD diagnosis, *n* = 6), AUC = 0.905.

The results of each of the different subtests will now be considered.

### 3.2. Results of the Nonword Repetition Task

For the assessment of validity and discriminative potential of the NWRT, the sum of all nonwords correctly repeated (0 = incorrect, 1 = correct) across the categories (LS Italian, LS Spanish and LU) was used, see [12]. This score was significantly associated with the BVL nonword repetition score (*n* = 36, *rho* = 0.776, *p* < 0.001). The discriminative potential of the NWRT was assessed with an exploratory analysis of diagnostic accuracy

(sensitivity and specificity analysis) with respect to the identification of the presence of phonological risk (vs. no phonological risk). The ROC curve (see Figure 6) based on the NWRT score produced an area under the curve (AUC) of 0.918 ($p < 0.001$). A cut-off point of two correct items would allow for a sensitivity of 0.684 and a specificity of 1.00 ($J = 0.684$), while a cut-off of three would correspond to a sensitivity of 0.842 and a specificity of 0.765.

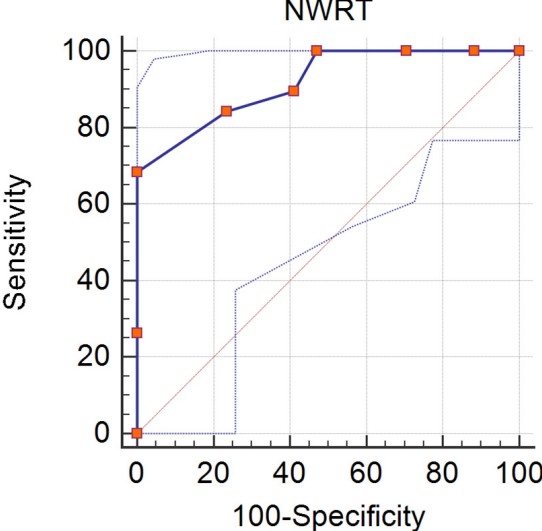

**Figure 6.** ROC curve for NWRT (distinction between phonological risk, $n = 19$, and no risk, $n = 15$), AUC = 0.918.

The NWRT score was significantly associated with the global ($n = 34$, *rho* = 0.551, $p = 0.001$) and risk score in the QUIR-DC ($n = 34$, *rho* = −0.554, $p = 0.001$) and also with the judgement of children's language skills by the SLTs and kindergarten teachers ($n = 36$, *rho* = −0.801, $p > 0.001$). Despite the small amount of nonwords selected for the NWRT implemented on the MuLiMi screening platform, for an exploration of construct validity, the association of children's repetition performance on LS IT and LS SP NWs was assessed and a significant correlation was found ($n = 36$, *rho* = 0.552, $p = 0.001$).

*3.3. Results of the Cross-Linguistic Lexical Tasks*

As recommended in the context of language assessments for multilingual children, only the results of the Italian screening task on verb comprehension were compared to the Italian standardised TFL. Children's performance in the Italian subtest of the CLT verb comprehension task was significantly correlated with performance in the TFL (raw scores, $n = 28$, *rho* = 0.779, $p < 0.001$).

To investigate the discriminative potential of the Italian subtest, an exploratory analysis of diagnostic accuracy was run with respect to the identification of the presence of lexical risk (vs. no lexical risk). The ROC curve (see Figure 7) produced an AUC of 0.826 ($p = 0.003$). A cut-off point of 24 correct items in the Italian CLT verb comprehension task would allow for a sensitivity of 0.800 and a specificity of 0.783 in identifying lexical risk ($J = 0.583$).

As to convergent validity, the Italian CLT verb comprehension subtest (raw scores) was significantly associated with the QUIR-DC total ($n = 26$, *rho* = 0.466, $p = 0.016$) and risk score ($n = 26$, *rho* = −0.431, $p = 0.028$). Similar results were obtained comparing the Spanish CLT verb comprehension score to the QUIR-DC total ($n = 26$, *rho* = 0.381, $p = 0.050$) and risk score ($n = 26$, *rho* = −0.422, $p = 0.028$). Additionally, the judgement of children's language skills by SLTs and kindergarten teachers was significantly associated both with the Italian ($n = 28$, *rho* = −0.425, $p = 0.024$) and the Spanish CLT verb comprehension task ($n = 29$, *rho* = −0.378, $p = 0.043$). The score (amount of verbs correctly identified) in the

Spanish version of this task correlated significantly with the score in the Italian version ($n = 25$, *rho* $= 0.656$, $p < 0.001$).

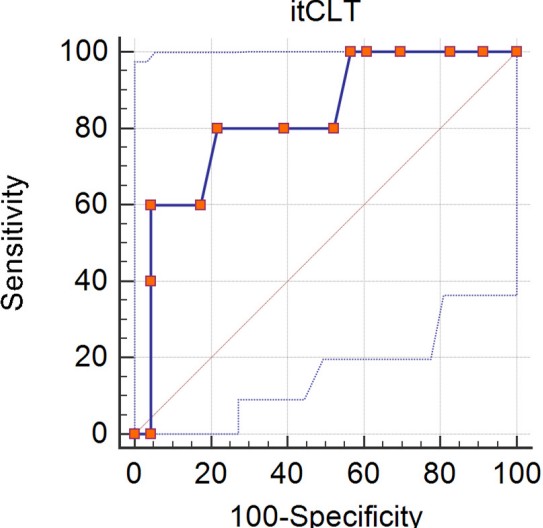

**Figure 7.** ROC curve for Italian CLT verb comprehension subtest (distinction between lexical risk, $n = 5$, and no risk, $n = 23$), AUC = 0.826.

*3.4. Results of the Subject-Verb Agreement Tasks*

In addition, for the assessment of validity of the screening task on subject–verb agreement, the results of the Italian screening task were compared to the Italian subtests on grammatical skills from the BVL. The Italian version of the WSIR SVA screening task significantly correlated with the three BVL scores ($n = 33$, $r_s$ ranging from 0.445 to 557, $p_s < 0.009$).

The discriminative potential of the Italian WSIR screening task on SVA was assessed with respect to the presence of morphosyntactic risk; the ROC curve (see Figure 8) indicates an AUC value of 0.789 ($p = 0.001$). A cut-off point of nine correct items in the Italian WSIR SVA screening task would allow for a sensitivity of 0.714 and a specificity of 0.842 in identifying morphosyntactic risk ($J = 0.556$).

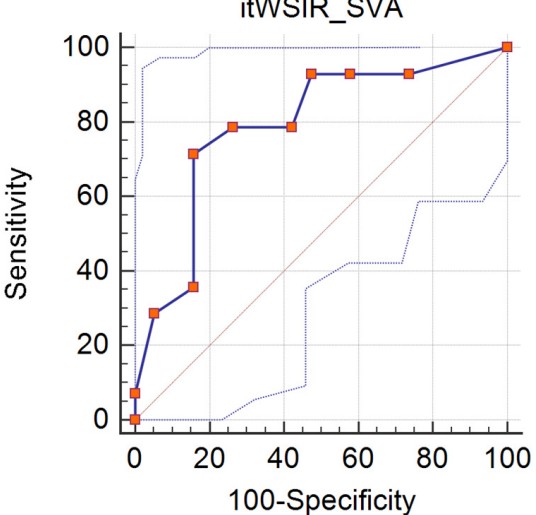

**Figure 8.** ROC curve for Italian WSIR SVA screening task (distinction between morphosyntactic risk, $n = 14$, and no risk, $n = 19$), AUC = 0.789.

Comparing the children's performance in the WSIR SVA screening task to the QUIR-DC, significant correlations were found with the global ($n = 31$, $rho = 0.593$, $p < 0.001$) and risk score ($n = 31$, $rho = -0.567$, $p = 0.001$). Although the Spanish version of this task was not correlated with the QUIR-DC scores ($p_s > 0.05$), the judgements of the SLTs and kindergarten teachers did correlate with both the Italian ($n = 33$, $rho = -0.610$, $p < 0.001$) and the Spanish version ($n = 33$, $rho = -0.356$, $p = 0.042$). A significant (but rather low) correlation between the two language versions of this task was found ($n = 31$, $rho = 0.364$, $p = 0.044$).

### 3.5. Results of the Finiteness Tasks

In addition, when comparing children's performance in the Italian finiteness WSIR task to their performance in the three subtests selected from the BVL, significant associations emerged (raw scores, $n = 28$, $r_s$ ranging from 0.447 to 0.680, $p_s < 0.017$).

Again, the discriminative potential of the Italian WSIR screening task on finiteness was assessed concerning the identification of the presence of morphosyntactic risk (vs. no morphosyntactic risk), the ROC curve (see Figure 9) indicates an AUC value of 0.900 ($p < 0.001$). For this task, a cut-off point of four correct items would allow for a sensitivity of 0.800 and a specificity of 0.889 in identifying morphosyntactic risk ($J = 0.689$).

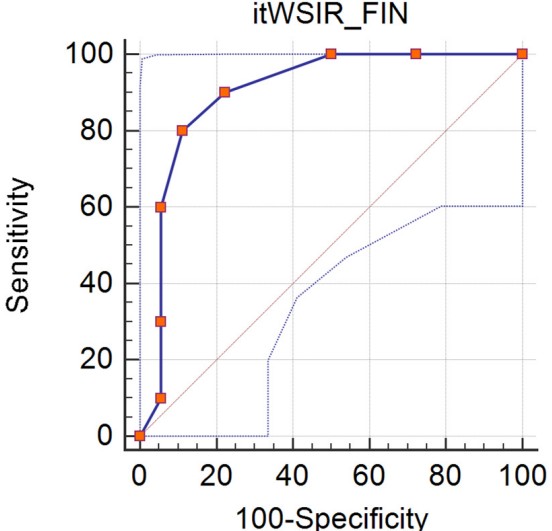

**Figure 9.** ROC curve for Italian WSIR finiteness screening task (distinction between morphosyntactic risk, $n = 10$, and no risk, $n = 18$), AUC = 0.900.

The Italian version of the WSIR finiteness task was significantly associated with the QUIR-DC global ($n = 26$, $rho = 0.472$, $p = 0.015$) but not with the risk score ($p > 0.05$). The Spanish version of the finiteness task correlated with both the QUIR-DC global ($n = 26$, $rho = 0.432$, $p = 0.028$) and risk score ($n = 26$, $rho = -0.411$, $p = 0.037$). The judgment of children's language skills by SLTs and kindergarten teachers was significantly associated with the Italian WSIR finiteness task only ($n = 28$, $rho = -0.494$, $p = 0.008$). Again, the two language versions correlated significantly ($n = 27$, $rho = 0.604$, $p = 0.001$).

### 3.6. Results of the Dynamic Novel Word Learning Task

Considering the small number of valid data on this task (due to various subsequent adjustments to the graphical presentation of the task that had to be introduced after data collection had started), no statistical analysis of the data was performed. The task will be the object of more extensive, dedicated studies and separate publications (in preparation).

### 4. Discussion

Based on data collected from children with, at risk of, and without DLD attending kindergartens in Italy, the computerised Spanish–Italian MuLiMi DLD screening that

automatically assesses children's language performance in both languages spoken appears to contribute to the risk identification of DLD. This has been shown by the correlations found between screening task performance and SLT, caregiver and kindergarten teacher questionnaires, as well as between screening tasks and standardised tests (convergent validity, Research goal 1). Furthermore, significant associations between standardised tests and screening tasks declared to measure the same skills indicate the suitability of the task paradigms and items applied. Performance on various screening tasks assessing the same linguistic areas in the two different languages are correlated with each other, indicating adequate construct validity. Even if no information on validity and reliability of the DNWL test can be obtained from our data due to an insufficient sample size, the available tests conducted with bilingual children show that it is possible to administer and evaluate (consolidation and testing phase) this task automatically [69].

Across screening tasks, children with a formal DLD diagnosis or with pathological scores in standardised tests indicating phonological, morphosyntactic, or lexical risk, underperformed compared to children with no pathological scores. This result indicates the discriminative potential of the tests (Research goal 2). Domain-specific risk scores were used for the exploration of diagnostic accuracy (specificity and sensitivity) of the screening tasks and yielded rather satisfying indices. These screening tasks could thus be used for the identification of possible problems across domains. Hence, the data preliminarily confirm that nonword repetition [52,56], grammaticality judgement (WSIR) of subject-verb agreement [58], and finiteness [64] can be considered useful clinical markers. In addition to the discriminative capacity of single screening tasks, the overall scores throughout the whole screening battery were also found to have good discriminative capacity. The results of the present study indicate that the language-specific as well as bilingual overall scores represent a meaningful selection and combination of tasks that discriminates between children with vs. without DLD (risk). These results obtained in a rather limited sample need to be confirmed on larger samples in the future. Crucially, our data indicate that it is possible to assess both languages spoken by the child [33,34] even though the person who administers the screening only masters one of them.

Besides the assessment of the suitability of the Spanish–Italian screening for DLD risk identification, the large amount of children who participated in this study and managed to carry out all of the screening tasks indicates that generally the child's interaction with the online screening tool is (a) feasible and (b) motivating for children of this age, see [77]. This is of particular importance in the light of the remote modality in which this screening study was carried out in collaboration with SLT clinics and kindergartens. Thus, the high and consistent participation rate also indicates the appropriateness and feasibility of remote assessments for DLD risk identification in bilingual children attending kindergartens. Although even earlier identification of DLD in children is generally recommended, the use of remote, direct language assessment of bilingual children at younger ages through computerised procedures is likely to be problematic. Indeed, it should be considered that children need (a) to have had a certain exposure to both languages and (b) to have reached certain stages of cognitive and motor development to be able to process the tasks and to interact with the system. Future studies are needed to investigate whether parts of the screening could already contribute to risk identification in younger children.

A limitation of the present study is the length of administration of the battery, which can seem to be unsuitable to children of this young age. Indeed, a large number of items have been included at this stage, but item analysis and item selection is ongoing, to both improve the reliability of the screening tests and increase their applicability in clinical and educational contexts. A further limitation concerns the fact that the data collected on the children's DNWL task performance were too few to be analysed and interpreted. In future studies, the dataset will be enlarged through more extensive data collection. Future studies should also include a more detailed questionnaire to be filled in by the children's caregivers on the quantity and quality of language exposure and use across languages. This will allow investigation of the complex relationships between exposure variables and

language performance at both the receptive and the productive level. An interesting result in this respect is the finding that the children in the TD group in our sample had more exposure to Spanish than children in the DLD and the at-risk groups. This counter-intuitive result points to the reliability of the measures included in the screening, since no bias can be claimed to be present depending on the reduced exposure in children with poorer language performance. Moreover, it confirms that exposure to both L1 and L2 is far from having any detrimental effects on the children's language development, and rather seems to show positive effects. A further unexpected result concerns gender distribution, with a higher percentage of females in the DLD group compared to the other two groups. This unusual result may either depend on random factors or reflect possible cultural factors biasing the referral of female versus male children to clinical services for diagnosis in this particular population.

Finally, the special role of the at-risk group in the present study needs to be acknowledged. Indeed, it is not possible to know whether the below-cut off scores obtained by this group on standardised scores are a consequence of reduced exposure or of an undiagnosed language disorder. The most likely scenario is that the former hypothesis will be true for some of the children in this group, and the latter hypothesis will be true for the other children. Nonetheless, this is exactly the group of children for whom a bilingual screening test is more needed. For this reason, they were considered a crucial benchmark for the assessment of the screening battery. It should be further considered that DLD children might also have been misdiagnosed because of an over-reliance on monolingual tests and norms.

Multiple sources of information were used to ensure that children at risk of DLD or TD were identified based on the results from tests but also from parents' and teachers' questionnaires, which should be less influenced by language exposure. In addition, the bilingual overall score should compensate for possible imbalances in language exposure in L1 and L2. Indeed, the correlations found between the QUIR-DC questionnaire scores and most subtests of the screening platform confirm the validity of the assessment. Contrastingly, the lower correlations found with the Spanish part of the screening are likely to reflect the very variable amount of exposure to this language in the present sample. Finally, by performing different group-related analyses, in which the at-risk group is either merged with the DLD group to indicate the presence of (any) language difficulty or excluded from the sample, we tried to control for the confounding effects of both potential under- and overdiagnosis. Clearly, the evaluation and integration of all different types of information is a complex and challenging task that requires caution and careful analysis. The potential of dynamic assessment tasks as an even more exposure-independent and thus very promising task has been highlighted in the literature [42–46] and is being assessed. More specifically, the validity and reliability of its computerised implementation are currently under investigation.

## 5. Conclusions

The assessment of both languages of bilingual children for DLD can be achieved through the bilingual screening tasks implemented on the MuLiMi platform, hosting various testing paradigms for static and dynamic tasks. The positive, preliminary results need to be confirmed on larger samples of children. Still, the tasks implemented across linguistic areas were found to have good performances in identifying children with DLD or language difficulties. This confirms the potential of remote assessments for the automatic, efficient, and non-complicated identification of DLD risk in bilingual children.

**Author Contributions:** All co-authors contributed to the manuscript. Conceptualization, M.L.L. and M.E.; methodology, M.L.L.; software, M.S. and F.G.; screening task construction: M.E., G.C., S.C., and M.L.L.; validation, M.E., M.L.L., S.C. and M.A., formal Analysis, M.L.L. and M.A.; investigation, M.E., G.C., B.L., M.A. and S.G.; resources, M.E., F.G., D.S. and M.L.L.; data curation, M.A., M.E. and M.L.L.; writing—original draft preparation, M.E. and M.S.; visualization, M.S., M.E. and M.A.; supervision, M.L.L. and F.G.; project administration, M.L.L. and M.T.G.; funding acquisition, M.L.L. and M.T.G. All authors have read and agreed to the submitted version of the manuscript.

**Funding:** This project received funding from the European Union's Horizon 2020 programme for research and innovation under the Marie Skłodowska Curie Grant Agreement No. 765556 and by the Italian Ministry of Health, Grant RC2022 to Maria Luisa Lorusso.

**Institutional Review Board Statement:** The study was conducted in accordance with the Declaration of Helsinki, and the protocol was approved by the Ethics Committee of the Scientific Institute Eugenio Medea, scientific section of the association "La Nostra Famiglia" Prot. N. 43/19, 17 June 2019.

**Informed Consent Statement:** All the participating children's parents or legal tutors gave their informed consent before they participated in the study.

**Data Availability Statement:** The data have been deposited in the Zenodo repository (DOI:10.5281/zenodo.7555154) and they will be made available on written request to the corresponding author (under appropriate agreements). The data are not publicly available due to limitations from the Ethical Committee and privacy issues.

**Acknowledgments:** We would like to thank all the families and teachers who participated in the study, as well as the speech and language therapists and psychologists from "Unità ospedaliera complessa neuropsichiatria infantile e dell'adolescenza" (UOC NPIA, ASST Lariana), Como, Italy as well as Annalaura Filippo and Raffaella Pozzoli from "Scientific Institute IRCCS Eugenio Medea", Bosisio Parini, Lecco, Italy. We especially thank the directors, teachers, and staff from schools in Bergamo, Italy "I.C. Alberico da Rosciate", "I.C. Gabriele Camozzi", and "I.C. Edmondo De Amicis", in Vercana, Italy "I.C. Don Roberto Malgesini Gravedona ed Uniti", and in Como, Italy "Scuola dell'infanzia Orsoline Dedalo". Furthermore, we would like to thank Francesco Vona as well as and Lukasz Moskwa from "i3Lab" and their students, Manvi Aggarwal, Riley Towers, Mohsen Pourshirazi, Mahsa Sedghi, Chen Yupeng, Zhou Xin, Zhao Yun, and Mohammad Rahbari Solout from Politecnico Milan for their support in the development of the platform. We also wish to thank Sheila Keeshan for her helpful assistance in proofreading the manuscript. Finally, we would like to thank Daniela S. Avila-Varela, who supported the creation of the testing material as well as Ewa Haman, Magdalena Łuniewska, Maja Roch, Tanja Rinker, Claus Zinn, and Myriam Cantú Sánchez who generously shared their expertise and material on the LITMUS-CLT tasks.

**Conflicts of Interest:** The authors declare no conflict of interest. The funding parties had no role in the design of the study; in the collection, analyses, or interpretation of data; in the writing of the manuscript; or in the decision to publish the results.

## Appendix A

**Table A1.** Overview of NWs selected for the Spanish–Italian NWRT.

| NW Category | NWs Selected |
|---|---|
| LS IT | ['mudjo], [fɔl'da:na], [blan'djeza], [maŋke'tale] |
| NLS | [fulsɑmit], [melinɑk] |
| LS SP | ['xaɲo], ['nwelo], ['laxo], [ajukom'jon] |

**Table A2.** Overview of examples of WSIR tasks.

| Item Category | Introductory Sentence | SVA | FIN |
|---|---|---|---|
| Italian example correct | "Chi lo dice giusto?" [Who says it right?] | "Io metto la sciarpa." [I put on the scarf.] | "Lui corre veloce." [He runs fast.] |
| Italian example incorrect | | "Io mette* la sciarpa." [I puts* on the scarf.] | "Lui correre* veloce." [He runINF* fast.] |
| Spanish example correct | "¿Quién lo dice bien?" [Who says it right?] | "Él duerme mucho." [He sleeps a lot.] | "Ella cae siempre." [She falls always.] |
| Spanish example correct | | "Él duermen mucho." [He sleepPL* a lot.] | "Ella caer* siempre." [She fallINF* always.] |

\* indicates ungrammatical forms.

**Table A3.** Scoring system of children's responses in the DNWL naming phase.

| Score | Prime | Response |
|---|---|---|
| 1 point | no prime | target stimulus, or high overlap with target stimulus: minimum 3/5 phonemes—including all vowels |
| 0.5 points | initial sound prime | target stimulus, or high overlap with target stimulus: minimum 3/5 phonemes—including all vowels |
| 0 points | initial sound prime | no response, or response given differs from target stimulus in more than two out of five phonemes |

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
