# Peer review of "Remote Screening for Developmental Language Disorder in Bilingual Children: Preliminary Validation in Spanish–Italian Speaking Preschool Children"

_applsci, doi:10.3390/app13031442_

Round 1

Reviewer 1 Report

This current paper is very interesting, it examined the effectiveness of applying a remote assessment for DLD risk identification in bilingual kindergarteners by a battery of experiments and questionnaires. This paper is well written, the idea is very attractive, and the design is relatively clear, and the data is reliable, but it needs further modification.

Abstract: from line 22 to 25, the goal of the research should be clearly stated. In line 29 the other half of the second parenthesis should be added where reasonable.

 Introduction: in the introduction part, research background, a brief literature of the field with problems revealed as well as the research gap and study question should be provided.

 Literature review: the present 1.1 to 1.2 should be put in the literature review section and the reviewed part should be related to the current study. For example, line 53 to line 61 should be moved to the introduction part. In line 76-line 79, are the factors like family history of language impairment, male gender and low level of parental education or SES considered in the questionnaire or the tests? If yes, certain description should be made in the relevant parts of the articles. Besides, in the end of the literature review a brief comment or summary should be made, so that readers might understand what you are going to do next based on the review of the literature.

 Materials and methods: the writing style should be in consistent , for example, in line 180 nine should be changed in to 9 in order to keep consistency with other numbers. The title of 2.2 should be changed into MuliMi platform and the tasks and more information should be provided for the platform, so that the reliability and the practicality can be ensured. More attention should be paid to language use, for example, in line 246, should be “one of the tasks …and the other”, in line 249 “ for each of the language versions of this task”, should be for “either of the..”, because there are only two tasks. Some of the paragraphs are too long, for example, line 267 to line 306 can be divided into three or more parts according to different themes of contents. In line 328 to line 338, more information of respondents as well as how the questionnaire is designed, the reliability of it should be introduced in detail.

Research result:  3.1 NWRT in line 394 and CLTs in line 416 should be introduced in full term, not abbreviated. The font of the figures should be in consistency, for example, the font of figure 6 is bigger than others.

 Besides, more modifications should be made on the language in order to clearly express certain ideas

Reviewer 2 Report

Dear authors,

I read your article Remote screening for Developmental Language Disorder in bilingual children: preliminary validation in Spanish-Italian speaking preschool children with great interest. The paper is well written, the rationale and importance of the study are well defined and the study seems well embedded within existing literature and tools on this topic. I do have some concerns, however, on the analyses conducted and found the results section too difficult/complex too follow. 

The Results section is too difficult to follow because of the many outcomes and different subgroups reported. It is very confusing that each analysis in based on a different number of participants and on different subgroups. These many comparisons/outcomes are also problematic because they increase the chances of finding a statistically significant result. In my view the authors should control for this (for example using Bonferroni correction). 

Also, I have some concerns and questions for clarification on the “At-risk” group of children. When the authors describe this group of children in their Methods section it is not entirely clear to me whether the low cut-off is based on an overall language score on the standardized tests or whether these children were considered at risk for DLD if they scored 2 SD below the mean on individual subtests of the standardized tests. Also, I am not convinced that including an At-risk group benefits the main goal of the paper (i.e., validating the tool in its assessment of DLD). One doesn’t know whether these “At risk” children eventually develop DLD or that their cut-off points indicate a language delay. As such, these children may confound the validation of the tool as screening tool for DLD, because we don’t know (yet) whether these children have DLD. 

Furthermore, given the main aims of the paper, the outcomes for the “overall screening score” seem most relevant/important. Therefore, I would suggest presenting the outcomes of the validity of the overall screening score first and then follow with an exploration of the validation for each of the individual subtests/linguistic areas. For all sections of the overall screening score, however, it is unclear why analyses are done on smaller subsets of the data.

Finally, to reduce the complexity of the results and the many comparisons I would suggest that the authors leave out the results regarding the DNWL test, particularly given the small sample that the outcomes of this task are eventually based on. In my view discussion and assessment of the validity of this task may be a study/paper on its own (with a larger sample size and more space to discuss the task in depth). 

Besides these concerns and questions, I would like to ask the authors to provide more details on the demographics of their participants: for example, it would be helpful to have more detailed information on the amount of exposure to Spanish of the individual children and whether his was linked to their status (TD, at-risk DLD, DLD). Given that DLD occurs in ± 5-7% of the children, the prevalence of DLD or at risk for DLD is relatively high in this sample (n = 27 which is 75% of the children). What are the genders of the children and how are the mean ages distributed over the different subgroups?
